# Chemical Sand Consolidation: From Polymers to Nanoparticles

**DOI:** 10.3390/polym12051069

**Published:** 2020-05-07

**Authors:** Fahd Saeed Alakbari, Mysara Eissa Mohyaldinn, Ali Samer Muhsan, Nurul Hasan, Tarek Ganat

**Affiliations:** 1Petroleum Engineering Department, Universiti Teknologi PETRONAS, Bandar Seri Iskandar 32610, Perak, Malaysia; fahd_19001032@utp.edu.my (F.S.A.); ali.samer@utp.edu.my (A.S.M.); tarekarbi.ganat@utp.edu.my (T.G.); 2Petroleum & Chemical Engineering, Universiti Teknologi Brunei, Gadong BE1410, Brunei; nurul.hasan@utb.edu.bn

**Keywords:** chemical sand consolidation, sand control, polymers, nanoparticles, epoxy, resin

## Abstract

The chemical sand consolidation methods involve pumping of chemical materials, like furan resin and silicate non-polymer materials into unconsolidated sandstone formations, in order to minimize sand production with the fluids produced from the hydrocarbon reservoirs. The injected chemical material, predominantly polymer, bonds sand grains together, lead to higher compressive strength of the rock. Hence, less amounts of sand particles are entrained in the produced fluids. However, the effect of this bonding may impose a negative impact on the formation productivity due to the reduction in rock permeability. Therefore, it is always essential to select a chemical material that can provide the highest possible compressive strength with minimum permeability reduction. This review article discusses the chemical materials used for sand consolidation and presents an in-depth evaluation between these materials to serve as a screening tool that can assist in the selection of chemical sand consolidation material, which in turn, helps optimize the sand control performance. The review paper also highlights the progressive improvement in chemical sand consolidation methods, from using different types of polymers to nanoparticles utilization, as well as track the impact of the improvement in sand consolidation efficiency and production performance. Based on this review, the nanoparticle-related martials are highly recommended to be applied as sand consolidation agents, due to their ability to generate acceptable rock strength with insignificant reduction in rock permeability.

## 1. Introduction

Approximately 70% of the total world’s petroleum are located in poorly consolidated reservoirs [1,2]. Over many years, producing sand with oil and gas from unconsolidated sandstone reservoirs has been a serious problem facing petroleum business [3,4]. The tendency and rate of sand production are influenced by many factors related to the degree of formation consolidation and flow characteristics. For instance, the decreasing pore pressure during the well life, resument of loose sand from the matrix. The main causes of sand production are the rate of reservoir fluids that creates frictional drag forces and a pressure that exceeds the formation strength. There is a critical flow rate above which sand is forced to be removed from the formation and entrained with the produced fluids. Rising water-cuts also affect sand production in a way that drops the relative permeability of oil over time after the production, thereby raising the pressure drop and inducing stresses, required to flow the well at the desired production rate, then yielding sand production [3]. Some researchers categorize the factors that affect sand production into three categories, namely driving factors, resisting factors and operational factors. The driving factors are forces that raise the movement of solids, like in-situ stress and its variation, saturation characteristics and fluid velocity. Resisting factors can be the strength of the materials, cohesion, friction, and the material grains and capillary action. Operational factors can be production rate and differential pressure [5,6], Figure 1. Other studies also disclosed that there are some critical parameters that can influence sand production, such as formation strength, permeability reduction and the reservoir fluids wettability, and mineral content [3].

Several researchers mentioned that the mechanism for sand failure can be shear, tensile, cohesive, pore collapse, and chemical effects that happen when cementation materials are weekend because of chemical interaction. Shear strength of a formation includes cohesion, which is the physical bonds between adjoining grains, and friction, which is contact forces between the grains [7]. The shear failure can be one of the most common mechanisms for sand failure. The shear failure happens when the shear stress at some plane inside the material overcomes the cohesion of the material and the internal friction. A tensile failure happens when the stress on a component exceeds the strength of the material [8]. Tensile failure can happen when the minimum principal stress ($\sigma_{3}$) becomes adequately negative, as illustrated in Equation (1):(1)$$\sigma_{3}\leq -T_{0}$$

$T_{0}$ is the tensile strength of the material.

When hydrodynamic erosion of the matrix takes place, sand grains start to release and move into the well. Sand arches and capillary forces can play a significant role in preventing sand production. Stable sand arches form when unconsolidated sand is pushed towards a small opening, like a perforation cavity or a slit in a screen. Capillary forces act as a stabilizing force for these arches [9,10]. The arches can fail because of the tensile failure criterion [5,11]. Sand production can occur because of reservoir formation damage. The reservoir formation damage can be categorized as natural damage, like fines migration, inorganic and organic deposits and swelling clay, pseudo-damage, namely (inappropriate installation of completion equipment and induced damage, such as wettability changes, chemical reactions, and incompatibility with completion fluids or drilling fluids and plugging from particles [3]. Unconsolidated sand formation can occur because of relatively young geological age and depositional environment no cement between the rock grain to attach them together by mineral deposition [12]. Nevertheless, sand formation consolidation occurs when the natural bonds attach the grains by cementing materials in the formation [13].

Sand production induces some problems, which includes downhole or wellhead equipment erosion and plugging, and subsidence of formation rock and walls demolition. In addition, it could also lead to a decrease in reservoir recovery, heightened maintenance cost, stimulation in the operational aspects problems and equipment damages, as well as other problems related to health, safety and the environment [14,15], Figure 1. Sand production constitutes erosion in surface and downhole equipment, i.e., growth the workover jobs. Sand production accumulate in downhole and surface equipment to rise the time and cost for cleaning. Sand production can plug the perforations to decrease the efficiency of production. Sand production decreases permeability and increases the pressure drop in the perforation tunnel because of void formation around perforation tunnel during sand production [3].

There are many techniques that can be used in the petroleum industry to manage and control sand production. These techniques include but not limited to production rate control, mechanical methods such as stand-alone screen, or gravel pack and chemical methods like resins [3,16]. Another method to control sand production is used to run the well on the maximum sand free rate (MSFR) [17]. In general, the mechanical techniques are more costly and time-consuming comparing to the chemical methods [17]. They have some other problems such as the decrease in productivity index, complexity of the installation equipment, damage during installation, workover issues, zonal isolation difficulties, screen collapse and plugging and erosion of equipment [3]. They are also ineffective in controlling fine sand particles. In addition, mechanical methods can interfere with the different type of completions and workover operations [18].

Chemical methods on the other hand can be an alternative approach to control the unconsolidated formation that produces a myriad of sand [11]. The chemical sand consolidation methods, (namely plastics, have been used to control sand production since early 1940 [19]. Chemical sand consolidation methods involve injecting reactive chemicals down to a target loose sand formation to create a bonding force between the loose sand grains, Figure 2.

These methods can be used to make up the formation matrix, increase the uniaxial compressive strength (UCS) of the sand formation to withstand the drag forces and maintain the sand formation permeability [20]. The chemical methods can also be used to decrease “the effect of tensile failure and hydrodynamic erosion of the formation matrix” [11,21,22], as well as in multiple zone completion [3]. Non-traditional chemical materials like polymers have been developed to enhance the physical properties of the materials [23,24]. Many researchers used polymers to control the sand formation because the viscosity of the polymers are not excessive, polymers can wet the formation solids, polymers provide sufficient compressive strength to consolidate the sand formation, polymerization time can be controlled, polymers can be applied in perforation, polymers are relatively low cost and some polymers can be used in high temperatures. The water-soluble polymers and microgels have a strong tendency to adsorb on the surface of the rock, and hence, provide a protective film to control erosion [25]. Overall, polymers can effectively improve the strength behaviors of the treated sand [26].

The aim of this article, however, is to disclose an up-to-date overview of the chemical sand consolidation techniques by highlighting and deliberating the most popular chemical methods and the latest advanced approaches, such as shape memory polymers and nanoparticles and their compounds. The present paper also discusses the performance of the chemical sand consolidation methods, with focusing on polymers, at their expected application conditions.

## 2. Phenol-Formaldehyde Resin and Urea-Formaldehyde Resin

Aldehyde condensation polymer can be obtained from condensation reaction of aldehyde organic compounds, Figure 3. Aldehydes undertake a broad array of chemical reactions, including polymerization. The aldehyde condensation polymers may be generated as a result of the contamination of aldehydes with specific types of molecules. In almost all cases, the employed aldehyde is formaldehyde, a highly reactive gas results from polymerization with phenol, urea, or melamine. The final product of this process is a series of important synthetic resins. The reactions ended with forming such polymers are called condensation reactions because of the associated water and other by-products released from the process. The resultant polymers can be called as phenol-formaldehyde resin, urea-formaldehyde resin, and melamine-formaldehyde resin. Examples of this type of polymers are Formaldehyde (HCHO) and Phenol (C_6_H_5_OH). Urea and melamine can also be called carbamide and cyanuramide, respectively which are crystalline solids including an amino group (NH_2_). 

The (phenol-formaldehyde) resin (plastic) has been studied for application in sand consolidations for more than five decades. The phenol-formaldehyde resin is a resin found by the reaction of phenol with formaldehyde as shown in Figure 3. It is the first synthetic polymer. The phenol-formaldehyde resins can be produced by two methods. The first method is that phenol and formaldehyde react to obtain a thermosetting network polymer. The second method restricts the formaldehyde to obtain a prepolymer, known as Novolac that can be molded and then cured with the addition of more formaldehyde and heat. The thermosetting network polymer results in chemical reactions that produce wide cross-linking between polymer chains to obtain “an infusible and insoluble polymer network” [27]. The flash point and boiling point of phenol-formaldehyde resin are 72.5 °C, and 181.8 °C, respectively. Phenol-formaldehyde resin has enhanced strength and dimensional stability. It also has significant dimensional stability on heating up to about 149 °C. Similarly, it has enhanced impact resistance and cast resins. The viscosity, the density at 25 °C and pH of formaldehyde and phenol resin are (100–150) cP, (1.16–1.21) g/cc and (9.1 or 10.1), respectively, [28]. The urea-formaldehyde (UF) polymer can be obtained by condensation polymerization of urea and formaldehyde. The properties of the UF polymer, include fast curing rate, low cost, easily adaptable, colorless bond line and excellent wood adhesion capability [29].

As an application of phenol formaldehyde resin as a sand consolidation chemical, Spain [30] applied a base-catalyzed plastic resin for sand production treatment, (see Table 1). The sand production treatment procedure includes three stages. Initially, an agent can be used to flush the sand to enhance the adhesion of the sand grain surfaces. The second stage, the resin solution can be injected after one hour at the given formation temperature. The final phase, diesel fuel can be injected to assure that the well bore is properly cleaned of the given resin solution. The sand treatment following the procedure, described above, resulted in a compressive strength of 1000 psi but with a reduction of the permeability to 50% of its original value, at formation temperature of 38–95.5 °C [30]. Sanfilippo et al. [31] showed that sand with compressive strength of 147 psi can be stable and may not be eroded with normal drawdown stresses. Reducing half of the original permeability makes the application of this plastic resin method is limited in oil industry. Talaghat et al. [32] studied the performance of six types of resin, including Novolak phenol-formaldehyde resin, resole phenol-formaldehyde resin, modified phenol-formaldehyde resin, two types of epoxy resins, and an acrylic resin to remedy the sand production in Ahwaz and Mansoori oil fields. They found that the best resin type is the modified phenol-formaldehyde resin. The experimental results of the modified phenol-formaldehyde resin showed that the permeability, the porosity and the compressive strength were 1500 and 3500 mD, 38–68% and 3000 psi, respectively at 29.5 °C, and 94 °C, Table 1 [32]. Mishra and Ojha [33] have applied a mixture of organic resins with inorganic resin and curing agents, in order to consolidate loose sand formation in oil fields. The organic resins are urea-formaldehyde (UF) and melamine-formaldehyde (MF). The inorganic resin is potassium silicate. Curing agents are ammonium chloride [AC](NH4CL), aluminum sulfate [AS] [A $l_{2}\;$(SO4${)}_{3}$] and sodiumbi-carbonate [SBC](NaHCO3). They pointed out that the inorganic silicate is less costly and available with the amino resins and displayed much enhanced uniaxial compressive strength more than 1300 psi [33]. Some pros and cons of phenol-formaldehyde resin and urea-formaldehyde resin methods as reported in literature are summarized in Table 2.

## 3. Furan Resin

Furthermore, some researchers studied other types of resins, like furan resin as shown in Table 3, to control sand in oil fields. Furan can be a heterocyclic organic compound, which contains a five-membered aromatic ring with one oxygen and four carbon atoms. Furan resin could be obtained from diverse furan compounds like furfuryl alcohol and furfural as illustrated Figure 4 [37]. The properties of furfuryl alcohol resin are 170 °C boiling point, 1.1285 g/cc density, 29 °C melting point, 359 °C critical temperature, 776 psi critical pressure and (5–8) pH. The viscosity and flash point of the furfuryl alcohol resin are (100–300) cP at 25 °C, and 65 °C, respectively [38].

Young [39] tried the application of furan for sand consolidation using a consolidating fluid composing of the furan resin (furfuryl alcohol resin), furfuryl alcohol solvent, silane bonding agent, surfactant and fresh water. The furan resin was used to bind unconsolidated sand formations. The silane bonding agent (silane coupling agent) enhances the adhesion of the resin to the sand grains. Silane coupling agents contain organic polymers and inorganic fillers, (namely glass, minerals, and metals). Silane coupling agents comprise inorganic reactive groups on silicon and can bond well with the most inorganic substrates, especially if the substrate comprises silicon in its structure. The furfuryl alcohol solvent was added to achieve some functions, like decreasing the viscosity of the resin, coating the formation solids, and to assist the sand consolidation at elevated temperatures. It also assists the clay treating chemicals to act as clay shrinking chemicals. This system has an apparent ability to permanently shrink certain types of hydrated clays within a sand formation to increase permeability. Fresh water acts as an additional ionization medium. It is used to speed up and activate the catalyst and the bonding agent. Therefore, it provides a faster resin curing and stronger consolidation. The surfactant acts as a demulsifier to break or prevent water blocks and emulsions. Young applied multistage of spacer solution and catalyst solution in this sand consolidation method. The spacer solution composes of diesel oil with small surfactant. The catalyst solution contains a strong organic acid dissolved in diesel oil which contains a surfactant. The spacer solution and catalyst solution can be used to spread the resin consolidating fluid. This sand consolidation system was experimentally investigated at temperature values of 15.5 to 149 °C, where the investigation results revealed that the compressive strength of the consolidated sand at the measurement temperature can reach up to 3000 psi. However, this proposed method is associated with some challenges, including multistage process, low flash point of the resin and high required pumps for the fluid [39]. Young [40] used chemical solution, (namely diesel oil as preflush solution) for the sand, resin composition comprises of Urea formaldehyde (UF-85) concentrate, furfuryl alcohol, urea, gamma-aminopropyltriethoxysilane and water and overflush (post flush) solution includes diesel oil and benzortrichloride to drop sand production in the oil industry. The compressive strength was displayed 1072 psi and the temperature conditions were 27, 60, 65.5 and 177 °C [40]. In another study, Young [41] proposed consolidating fluid and a low viscosity aqueous fluid to control sand formation. The consolidated fluid comprises a mixing of terahydrofurfuryl methacrylate and a furan resin. the low viscosity aqueous fluid contains of potassium chloride, sodium chlorides, ammonium chloride, calcium-calcium bromides, oil field brines and sea water. Additionally, a diluent resin additive, (namely astetrahydrofurfuryl acrylate), was added to the given consolidated fluid. The chemicals like an acid curable resin, viz. sulfamic acid, and a delayed acting catalyst which contain of the reaction product of an acid and an alkali metal or ammonium molybdate. The experimental results of this furan resin method showed that the compressive strength was (112–1170) psi at (29–57) °C [41]. Friedman et al. [21] used a polymerizable resin like an oligomer of (furfuryl alcohol) furan resin, a catalyst, organic diluent and ester. The catalyst was used for the polymerization of the monomer or oligomer. The ester, (namely ethyl or butyl acetate), was injected into the formation and forced to enter the void space in the formation for consolidating the sand formation. Prior to the treatment solution, a preflush solution, (namely ester (ethyl or butyl acetate)), and an acid catalyst (sulfuric acid) were injected into the borehole to remove water from the pore spaces of the treated formation and to remove oily residue from the grains. The polymerization happens at high temperature (150 °C) for a few seconds. However, at the surface conditions (30 °C) the mixture of monomer and acid are not reactive for some days. The well could be shut in from one to ten days and preferably at least one week to allow the chemical materials to react with the sand in the well. The weaknesses of this method are that some environmental unfriendly materials like sulfuric acid are used as additives to the consolidation solution. The performance of the catalyst depends on the temperature which must be less than 60 °C. Moreover, the multi-fluid injection procedure needs more time and it is associated with higher cost as compare to a single fluid injection. The three-system process has a negative effect in maintaining the sweep efficiency of each fluid through the porous media [21]. Fader et al. [42] studied a modified furan resin chemical system comprises of furfuryl alcohol with ethylacetate, methanol and sulfuric acid to consolidate a sand formation in Kern River field, CA, USA. Experimental investigation conducted on the modified furan resin revealed that the compressive strength can reach up to 550 psi, while there is 10% permeability reduction. It is reported by researcher that the modified furan resin method can be applied at temperatures (15.5–160) °C, [42]. Parlar et al. [43] experimentally investigated furan resin composition which encompasses solvents, oil wetting agent and nonemulsifier and coupling agent to consolidate sand formation. This furan resin method showed that the maximum unconfined compressive strength is more than 2000 psi and the retained permeability is more than 100% for clays, 68% for clean sand with the average permeability (80–85)% at measurement temperatures of 38 °C, 66 °C, 93 °C and 121 °C. The limitation of the proposed furan resin method is that the placement of the chemical through all perforations is critical [43]. Todd et al. [44] employed enhanced hardenable resin compositions to consolidate an unconsolidated sand formation. The enhanced resin compositions contain an organic resin for hardening (namely furan resin), an aminosilane resin to act as a solid coupling agent, a viscosity breaker and a surface active agent. The solid coupling agent is used to break separating films of viscous carrier fluid between solid coupling agent and the surface active agent. The aminosilane coupling agent can be 3-aminopropyltrimethoxysilane and the surface active agent can be ethoxylated nonyl phenol phosphate ester. The employed enhanced hardenable resin compositions further contain a viscosity reducing diluent such as methanol and butyl alcohol. The viscous carrier fluid breaker can be organic peroxides and/or a hydrolyzable ester. They measured compressive strength of the sand consolidation after the treatment was found to be as high as 3000 psi at 93 °C [44]. Appah [45] used furan-phenolic resin to consolidate the sand formation. In addition, he proved that the compressive strength was 3000 psi for the said consolidation method [45]. Nguyen [46] used resin composition to consolidate sand formation and to fracture the producing zone. The method is processed in three stages. The first phase formed perforations through the casing and cement adjacent to the treatment production zone in the wellbore. The second stage is that a furfuryl alcohol resin, furfuryl alcohol, a silane coupling agent and a solvent for the given resin are injected through the perforations to the target formation. In the third stage, a fracturing fluid was immediately injected into the production zone at an adequate rate and pressure to fracture the target zone. The fracturing fluid includes proppant particles coated with a second hardenable resin composition. The proppant particles are graded sand. The second hardenable resin composition includes a furfuryl alcohol resin, a furfuryl alcohol, a silane coupling agent, a solvent for the resin and a surfactant. The function of surfactant is to facilitate the coating of the given resin on the given proppant particles and to cause the said hardenable resin to flow to contact points between adjacent resin coated proppant particles. The second resin composition further contains a hydrolyzable ester to break gelled fracturing fluid films on the proppant particles. The hydrolyzable ester includes a mix of dimethylglutarate, dimethylsuccinate and dimethyladipate. Nguyen investigated the sand consolidation method at 27 °C and less than 149 °C. Moreover, he outlined that the compressive strength ranges from a minimum value of 80 psi to a maximum value of 1642 psi after 7 days [46]. Some benefits and drawbacks of furan resins as reported in literature are summarized in Table 4.

## 4. Epoxy Resin

Epoxy resin can also be called polyepoxides and can be a class of reactive polymers and prepolymers, which comprise epoxide groups, as shown in Figure 5. The reaction of epoxy with polyfunctional hardeners or with themselves can form thermosetting polymer. Epoxy resins have substantial mechanical strength, high resistance to heat, outstanding resistance to chemical and substantial adhesive strength [48]. Epoxy resin can be used in water shut-off treatment [49]. Epoxy resins as liquid can be toxic to aquatic organisms and are irritant to the eyes and skin [50]. The properties of epoxy resin are 1.15 g/cc density, 150 °C boiling point, 164 °C flash point and (200–1500) cP viscosity. Epoxy can be capable of practical operating at (−250 °C to 120 °C) and can be capable with higher temperatures (215 °C) for short time periods only (3000 h.) [51].

There are several studies in the literature on the application of epoxy resin for sand consolidation as demonstrated in Table 5. Penberthy et al. [52] used an aqueous organic preflush and epoxy resin for sand consolidation. To improve the performance of the sand consolidation, initially the resin must be wet and adhered to the surface of the sand grains. The sand grains in many reservoirs can be water wet formerly, it is difficult for the resin to replace the water on the surface of the grains. Thus, they applied ethylene glycol isopropylether solvent and isopropyl alcohol (IPA) with ethoxylatedhexyl alcohol solvent as a preflush to remove the water in the reservoir. The treatment process results in 2000 psi compressive strength [52]. Dees et al. [53] used epoxy resin consolidation and explosive for treating unconsolidated sand formation. The explosion can be achieved using a cord holding suitable explosive materials. It is used to divert the chemical solutions through all perforations in the casing of the well and to increase the production rate. After the explosion is processed, the consolidation method further utilizes a perforating gun to form addition perforations in the casing. The diverting agent can be added to the chemical solution of the said resin composition to displace the chemical solutions into the target zone. The compressive strength and permeability to the original were obtained as 5000+ psi, and 67%, respectively at the temperature range of 38–104 °C [53]. Dees [54] used an epoxy resin and a gas generator to control the unconsolidated sand formations. The gas generator was used with the resin fluid to increase the flow velocity towards the given production zone. The gas generator contains of a propellant which is a modified nitrocellulose. After the fluid epoxy resin with gas were injected, the flush solution was pressurized to assist the formation of the consolidation porous permeable matrix. The flush solution can be a hydrochloric acid solution. Dees [54] determined the compressive strength as 7000 psi and the permeability to original is 50% at 10 °C to 121 °C [54]. Dewprashad et al. [55] practiced a new HT epoxy resin, hydroxypropylguargum and KCl to consolidate a loose sand formation. The new HT epoxy resin can be used to treat fracturing fluids. Furthermore, this method does not affect the fluid pH and has less impact on gel breakers. The resin treatment takes adequate time to become hardened because of a lower cure rate at high temperature, so the consolidation operation can become easier to palace the resin at the target sand zone. The experimental investigation, conducted by the researchers, showed that the compressive strength was 1340 psi at temperature up to 204 °C [55]. Chaloupka et al. [56] applied catalyzed epoxy fluid to consolidate a sand formation in the Mahakam river delta, Indonesia. They reported that the regained permeability is 63% at the temperature range of 93–177 °C, [56]. Marfo et al. [13] used an internally catalyzed epoxy resin to consolidate sand formation in the Mahakam Delta, Indonesia. The internally catalyzed system comprises a hardener and accelerator. The catalyst can be added in the surface and the amount can be determined by the temperature of the formation to be consolidated. It is time-dependent with rapid reaction in high temperature formations. Externally catalyzed systems can be overflush. The overflush can be used after the resin is injected into the zone. The overflush may be aqueous or hydrocarbon solution, which contain a hardener or accelerator chemical introduced as a curing agent. They disclosed that the regained permeability was 63% and the compressive strengths were 57 psi and 235 psi for the treated sand without Bentonite clay, and with 1% of bentonite clay, respectively. The rise in compressive strength as the particle size drops because of rise in surface area of the smaller particle size [13]. Riyanto et al. [57] used a chemical consolidation composition composing of epoxy resin, coupling agent, accelerator and hardener. The accelerator was added to accelerate the curing process and shorten the well shut-in time. They used three phases to consolidate the sand formations. The first stage is preflush, which contains KCL and an aqueous base surface modification agent (ASMA). The second stage is the main aqueous based resin. The third stage is the post flush which is KCL. They investigated three samples with different aqueous based resin to find the compressive strength and regain permeability. The first sample composes of 3% KCL as preflush, aqueous based resin with accelerator (8PV) and 3% KCL (2PV) as post flush. The second contains 2% ASMA (2PV) and 3% KCL (3PV) as preflush, aqueous based resin with accelerator (8PV) and 3% KCL (3PV) as post flush. The third sample composes of 5% ASMA (2PV) and 3% KCL (3PV) as preflush, aqueous based resin with accelerator, foaming agent and N2 (8PV) and 3% KCL (3PV) as post flush. The unconfined compressive strength (UCS) ratio (top/bottom) were (284/103) = 2.76 for the first sample, (439/191) = 2.30 for the second sample and (797/148) = 5.39 for third sample. The regain permeabilities were (73, 75, almost 100)% for sample number 1, 2, 3 respectively. The condition of the sand consolidation measurements was at 63 °C. They applied this sand consolidation in Sarawak, East Malaysia and displayed that there was no sand production after the treatment. Moreover, the production rate was increased twice the production rate with other sand control method, (namely thru tubing screen) [57]. Hadi et al. [58] used an epoxy resin to consolidate a sand formation in Mahakam Delta, East Kalimantan, Indonesia. They showed that the compressive strength and the permeability were 614–1816 psi, and 100–1068 md, respectively [58]. Palencia et al. [14] applied an epoxy resin to control a sand formation. They mentioned that the compressive strength and the permeability were 217.21–694, and 8.15 × 10^−7^–1095.1 mD, respectively [14]. Alanqari et al. [59] used Bisphenol A/Epichorohydrin and Butyl glycidyl ether and cyclohxanedimethanol resins cured with polyoxypropylene diamine and diglycidylether of bisphenol-A epoxy resin to control the sand formation at Up to 146 °C. Al-Mulhem [60] utilized an epoxy resin and curing agent (Expedite A& B by Halliburton Energy Services) to consolidate the sand formation. They stated that a permeability to original was 70% [60]. Eluru et al. [61] used a sugar based hardenable resin and a hardening agent to make a resin compound to consolidate the sand formation. The sugar based hardenable resin is a glucofuranoside-based trifunctional epoxy resin and a glucofuranoside-based trifunctional epoxy resin, [61] (see Table 5). The main advantages and disadvantages of epoxy resin, as reported in literature, are summarized in Table 6.

## 5. Amino-Aldehyde Polymer

Amino-aldehyde polymer is the polymer, which includes amino group and formaldehyde. The amino-aldehyde polymer can be also acquired by condensation polymerization of amino and aldehyde. The properties of the amino-aldehyde polymer can be the ease accessibility and the low cost of their raw materials, (namely melamine and formaldehyde), the well combination of their oligomers with water and the easy transition of oligomer-polymer, not only at high temperature, but also at room a room temperature. However, the amino aldehyde resin has tremendously low durability [64]. Example of amino aldehyde (aminoacetaldehyde) is shown in Figure 6.

There were other chemical consolidation materials, such as Quasi Natural Consolidation (QNC) and a mixture of amino-aldehyde resin applied to control the sand formation, Table 7. Larsen et al. [65] studied QNC which comprises Ca^2+^, urease and urea. They found that calcium carbonate can form bridges between sand grains and can strengthens the loose sand pack. Experimental results depicted that the rock permeability decreased about 25% from the original (almost ten Darcy) and the compressive strength obtained was 1531 psi at temperature of 25–65 °C, [65]. Lahalih and Ghloum [66] applied polymeric materials which comprise a mixture of amino-aldehyde resin and water-soluble resin. Additionally, some additives can be added to the polymeric chemical method such as dispersants, inorganic and organic cross-linkers, (namely acetic acid and soil stabilizer). The polymeric method was tested without and with KCl brine. The polymeric method laboratory measurements offered that the compressive strength was 497 psi to 1422 psi at the compositions of (1.0, 0.4, 25) PV and at the temperatures of 60, 80, and 110 °C, respectively, [66]. The results of organic resins and inorganic resins showed that there was a drop in the values of porosity and permeability as the amount of the binding agent and the amount of reactive component were raised [67]. Some benefits and drawbacks of amino-aldehyde polymers, as reported in literature are summarized in Table 8.

## 6. Silicate Polymer

Silicate is any member of a family of anions that contains silicon and oxygen, Figure 7. Silicate anions are often large polymeric molecules with an extensive variety of structures, containing chains and rings, [71]. Silicates are found in a variety of structures. Many of them are identified from their crystalline state and are in a highly polymeric forms [72]. Organic silicates shows excessive differences in physical properties, depending primarily on the position of the metal and on the alkyl group [73].

As an application to oxidation, Anthony [74] used organic silicate, alcohol and water for hydrolyzing and forming a coating-like binding agent which could consolidate and strengthen unconsolidated sand formation in two steps, Table 9. Firstly, organic silicate and alcohol are injected into the target formation followed by water. The water can cause the organic silicate to hydrolyze and polymerize. He utilized the proton source water which contains a weak base and a strong acid, where the strong acids are unsafe aspects to be used in the treatment. Some chemical materials may be sensitive to formation sand and fluid type. The alcohol-silicate solution reacts with residual formation water, present in less mobile areas of the formation, not swept by both alcohol and silicate solution and water. Therefore, it is needed to preflush the target formation with a liquid, such as alcohol that will totally displace the formation water. Alcohol has low surface tension reaction with water, so it is preferable preflush for the formation. He concluded that it is necessary to use new consolidating materials, which are safer and less sensitive to formation and sand fluid, to avoid the problems of safety issues and the negative effect on the formation [74]. Shu [75] applied three phases to consolidate the sand formation. Initially, an aqueous silicate solution was injected into the production zone through the perforated casing. After that, a water-immiscible hydrocarbonaceous liquid as a spacer volume was injected to the zone. Finally, a water-miscible organic solvent including inorganic salt and an alkylpolysilicate was injected into the zone. The aqueous silicate solution is an ammonium silicate solution. The rock that had undergone aqueous silicate solution consolidation was examined at 79.5 °C, and the retention permeability is high without a specific value of permeability retention [75]. Dwivedi and Singh [76] analyzed non-polymeric materials, such as sodium silicate. They reported that the compressive strength resulted from the consolidation is in the range of 0–398 psi at temperature ranging from 20 to 60 °C. Cobianco et al. [77] used a three-stages process to control sand production in oil fields. Initially, 3% KCl was injected as a preflush to avert any contact between the formation water and the sodium silicate. The second stage, non-polymeric materials like sodium silicate solution was injected through the wellbore into the sand zone. In the same phase, the viscosity of sodium silicate solution was decreased by using a dilution fluid. A weak organic acid solution was injected in order to stabilize the amorphous silica network. The final stage, the nitrogen gas was injected as an overflush removing the extra of silicate solution from the porespaces and drying the sodium silicate coating sand grains and to promote sodium silicate salts precipitation. The experimental investigation on formation samples after the application of sodium silicate method disclosed that permeability retention was (70–80)% and the compressive strength was (284–427) psi at 60 °C [77]. Some advantages and disadvantages of silicate polymer methods as reported in literature are summarized in Table 10.

## 7. Oxidation and Hydrocarbon

The crude oil and oxygen were used to consolidate the sand formation. When the crude oil reacts with oxygen, an oxygenated hydrocarbon, coke-like material, aldehydes, alcohols can be obtained [79].

As an application to oxidation, Jennings et al. [80] used a method for heating an unconsolidated sand formation in a zone penetrated by a hydrocarbon-bearing reservoir. The method involves in-situ combustion of the formation to cause consolidation of the sandstone matrix, in which the consolidated sand acts as a natural and effective barrier to the passage of formation sand to the well-bore. A combustion-supporting oxygenated foam can be injected through a well into the unconsolidated sand formation. The foam can be made using any hydrocarbon product like diesel as the base fluid. Combustion can be initiated between the oxygenated foam and the hydrocarbons in the reservoir to burn the hydrocarbons in the unconsolidated sand, and create a sand consolidated zone that acts as a barrier between the reservoir and wellbore. The burning of the hydrocarbons can be increased by increasing the oxygen concentration. The combustion is initiated using a combustion igniter throughout the conduit from the surface into the target zone [80]. Barclay et al. have used an electrical igniter, while Shu has applied gas-fired burner [81,82]. The study of the oxygenated foam indicated that the compressive strength was 3000 psi and the permeability to the original was 70% at 138 °C, [80]. Aggour et al. [83] and Osman et al. [84] used low-temperature oxidation of crude oil in sand formation consolidation. They reported that the compressive strength following this process is in the range 375–1264 psi and the permeability loss was 22%, [83,84]. In another study, Aggour et al. [85] developed LTO method by using six fluids: Arabian Heavy Crude Oil, aged samples from the crude oil, refinery heavy residue, three different concentrations 40%, 60% and 80%, of asphalt-reformate solutions. They observed that the compressive strength was (1800–2300) psi and the retained permeability to the original was (86.4–95.5%) at (100–150) °C [85]. Khamatnurova [86] used a long chain hydrocarbon viscosifier, a curing agent and a thiol crosslinking agent to control the sand formation, (See Table 11). Some pros and cons of oxidation and hydrocarbon methods as reported in literature are summarized in Table 12.

## 8. Shape Memory Polymers

They are known as intelligent materials that can return from a deformed state temporary shape to their original permanent shape due to the effect of an external stimulus like temperature. They can be stimuli-responsive materials. Alloys, ceramics, polymers and gels can be applied to exhibit shape memory behavior [87,88,89]. Shape memory polymers can be molded into a cylindrical geometry with a pre-specified outer diameter (OD) and inner diameter (ID). The Shape memory polymers can go through a geometric alteration process to achieve the required run-in diameter. The Shape memory polymers have pore throat structures with diameter of 60–160 μm. This material can be used to permit all particles less than 43 microns to flow easily through the filter screen and stop the sand particles greater than 43 microns to control sand production. Carrejo et al. [25], Wang et al. [90], Leung et al. [91], and Fuxa et al. [92] used shape memory polymers to control the sand formation, (See Table 13). The used shape memory polymers have shown permeability ranges from 5 to 600 mD at temperature of 37.8–93.3 °C [25,91,92]. Some advantages and disadvantages of shape memory polymers methods as reported in literature are summarized in Table 14.

## 9. Hydrolysate or Precondensate Consolidation Agent

Also, some chemical materials such as a hydrolysate or precondensate were investigated to control the sand production in the petroleum industry. The hydrolysate or precondensate contains at least one organosilane and one hydrolysable silane. The organosilicon compound includes carbon silicon bonds C-Si. The word hydrolysis is derived from combining the word hydro, which means water, and the word lysis which means unbind. The hydrolysable silanes can be prepared by reacting an alkyl $Y-OC\left(O\right)-CHBr-CH_{2}Br\;$with an amine of the formula Z-NH_2_. Example of hydrolysable silanes is shown in Figure 8, [94].

As application of hydrolysate or precondensate consolidation agent, Endres et al. [95] used a consolidation agent which comprises a hydrolysate or precondensate to control sand production in oil industry, Table 15. They demonstrated that the application results in a compressive strength of 335 psi at 150 °C [95]. Some benefits and drawbacks of hydrolysate or precondensate consolidation agent method as reported in literature are summarized in Table 16.

## 10. Permeability Enhancing Additive (PEA)

Permeability enhancing additive (PEA) is a chemical material which acts as a diluter of the resin. The permeability enhancing additive (PEA) has two functions. The first function is to decrease the viscosity of the resin to be easily pumping into the wellbore. The second function can be used as the liquid phase to fill in the pore spaces of the consolidated sand formation to prevent the permeability reduction.

As examples of the implementation of permeability enhancing additive (PEA), Kalgaonkar et al. [97] have conducted experiments using resin composing of two chemical components (donated as A and B) assisted with permeability enhancing additive (PEA), Table 17. The first chemical component is the main resin chemical while the second component is the curing agent for the resin chemical, which is used to produce a low viscosity fluid for easy pumping. The optimum composition of the chemical components was found to be 1:1:0.8 (A: B: PEA). When investigating the resin with PEA they found that the consolidated sand pack yield strength is 4–6 times the compressive strength of the original core, with values exceeding 1000 psi. In addition, the new method has shown other attractive attributes, including easy mixing procedure, maintaining the permeability, and higher controllable curing time up to some days, so averting any premature setting of the resin in the wellbore [97]. Some advantages and disadvantages of permeability enhancing additive (PEA) method as reported in literature are summarized in Table 18.

## 11. Polyurethane Resins

Polyurethane resins can be formed by the polyols and organic polyisocryanates reaction, as shown in Figure 9. In addition, some additives can be added to the solution namely catalysts, foaming agents, surfactants and crosslinking agents. Polyurethane resins have the ability to create significant adhesion to base materials, [98,99].

As applications of polyurethane resins in sand consolidation, Spurlock et al. [100] applied kerosene or diesel oil and polyurethane solution to control the sand formation. They reported that the compressive strength after the application was 161–600 psi, while the permeability to original was 80% at 60 °C, [100]. Liu et al. [24,101] used poly-oxypropylene diol, poly-oxyethylene glycol, poly-caprolaclone glycol, and toluene to consolidate an unconsolidated sand formation, Table 19. They disclosed that the shear strength cohesion values for 50% polymer with densities of 1.40, 1.50 and 1.60 g/cm^2^ were 120.19, 140.28, and 204.22 kPa respectively, [24,101]. Some pros and cons of polyurethane resins methods, as reported in the literature and summarized in Table 20.

## 12. Polyacrylamide Polymer

The chemical sand consolidation methods discussed above have some unfavorite feature such as multiple phases for consolidating, environmental unfriendliness, less compressive strength and difficulty of surface pumping and handling. New developed chemical materials like polyacrylamide polymer were used for consolidating the grain sand of the formation. Polyacrylamide has a high molecular weight water soluble or swellable polymer. Polyacrylamide polymer can be a polymer, which can be obtained from acrylamide subunits, Figure 10. The acrylamide can be an organic compound with the chemical formula CH_2_=CHC(O)NH_2_. The properties of acrylamide are 1.322 g/cc density, 84.5 °C melting point and 138 °C flash point. The properties of polyacrylamide solution are 4–6 pH and 1.11 g/cc at 25 °C in 50 wt.% in H_2_O and (2–3) cP in 15% $H_{2}O$ [105].

As application of polyacrylamide polymer in the sand consolidation, Falk [106] has studied an aqueous solution comprises an acrylamide polymer, an aldehyde and a phenol to control sand production. The aqueous solution was applied at 50–100 °C [106]. Sydansk [107] evaluated an acrylamide-polymer/chromium (III)- carboxylate gel to matrix reservoirs as near-wellbore total shutoff treatments. This gel was applied at temperature up to 127 °C [107]. Mixing a polymer solution with a cross-linker which is called hydrogel can be used to control sand formation because it has adjustable viscosity, lower required concentration and good injectivity [108]. Salehi et al. [108] studied hydrolyzed acrylamido propyl sulfonated acid (polyacrylamide polymer sodium salt) and a cross-linker chromium triacetate to control sand production. Their study showed that the compressive strength was risen 30 times the original one and a growth in the polymer and the cross-linker provides an increase in the compressive strength of the remediation sand formation. The study of the polymer and the cross-linker demonstrated a positive impact on sand consolidation. However, there is also a lack of investigation in the sand consolidation mechanism, permeability measurements and the measurements were conducted at only 90 °C [108], Table 21. Some advantages and disadvantages of polyacrylamide polymer methods as reported in literature are summarized in Table 22.

## 13. Water-Based and Saline-Based

Some researchers used water-based and saline-based chemical materials to consolidate loose sand formation. Bhasker et al. [114] and Songire et al. [115] applied an aqueous-based epoxy resin to consolidate the sand formation. Othman et al. 2017 [116] utilized a solvent-based epoxy resin to control the sand formation in Peninsular Malaysia, Table 23. Shang [69] used a melamine formaldehyde resin water-based to dilute the solution to become less viscous and easier to pump. Furthermore, the melamine formaldehyde resin has an outstanding thermodynamic and mechanical properties, is not toxic, is more compatible with the fluid in the formation, has good compressive strength (742–911) psi and permeability (932 mD to 2736 mD) [69]. However, the compressive strength and permeability were measured at 60 °C only. Another limitation of the melamine formaldehyde resin is that the resin is expensive, [68]. Reddy et al. [117] investigated polymerizable aqueous consolidation composition for a formation sand control, Table 23. The polymerizable aqueous consolidation composition contains a water-based fluid, a polyvalent metal salt of a carboxylic. Additionally, polymerization agents, a water-soluble polymerization initiator, solids-free, aqueous flush fluid, a solvent, a non-ionic carboxylic acid derived comonomer, a coupling agent, a polymerization retarder and/or crosslinker can be added to the polymerizable aqueous consolidation to improve the polymerization and curing process. The polymerization agents are the chemical materials which can accelerate the rates of the polymerization aqueous consolidation. The water-soluble polymerization initiator can be used to initiate polymerization and curing of the polymerizable aqueous consolidation composition. The solids-free, aqueous flush fluid are utilized to remove the polymerizable aqueous consolidation composition from interstitial spaces between the coated, unconsolidated particulates or between the coated, proppant particulates in the proppant pack. A solvent, such as an alcohol can be added to the aqueous consolidation composition to aid in the solubility and/or polymerization and curing of the polyvalent metal salt of carboxylic acid. The non-ionic carboxylic acid derived comonomer can be added to the polymerizable aqueous consolidation composition to improve the polymerization. A coupling agent can help to bond, adsorb or/and consolidate materials which have different polarities. Polymerization retarders can be applied to slow down the polymerization reaction. A crosslinker can be utilized to help in the formation of a water insoluble three-dimensional polymer network when the water-soluble polymerization initiator composition starts polymerization [117]. George et al. [118] applied saline-based resin method to consolidate sand formation, Table 23. They disclosed that the benefits of saline-based resin method are safer, environmentally friendly, and low viscos (1 cP) (i.e., requiring less pumping pressure). Besides, a value of regained permeability is more than 80% while the unconfined compressive strength (UCS) is about 200 psi. In general, the saline-based resin method is operative to consolidate sand formations with clay (up to 15%), [118]. Nonetheless, the previous saline-based resin methods have less unconfined compressive strength, approximately (200–911) psi compare to the other methods. Some benefits and drawbacks of water-based and saline-based methods as reported in literature are summarized in Table 24.

## 14. Nanoparticles Materials

Recently, nanoparticles materials have been used intensively in the oil industry [119]. Some nanoparticles materials were used to consolidate the sand formation because they give new properties as well as change the properties of the resins [120]. Espin et al. [120] investigated the application of (1–200 nm) nanoparticles formed of molecules of organic, such as hydroxyls and inorganic like SiO_2_ silica to form a bond of enough strength with adjacent contacting grains whereby the sand formation is consolidated, Table 25. In accordance with the prior method, the component of inorganic has an attraction for the sand grains. In addition, the organic component permits polymerization bonding of the inorganic component to contact sand grains under non-acidic (basic pH) conditions. They showed that the nanoparticles materials can significantly improve formation strength without negatively impacting the permeability and porosity. They used aqueous suspension of nanoparticles and the water-based fluid was used as a displacing fluid to dislodge formation fluids into unconsolidated formation and away from the well. Mishra and Ojha [67] studied silicon dioxide nanoparticles (10–20 nm particle size) with urea formaldehyde resin [UF] to improve the capacity of urea formaldehyde polymer chain to bind the grains of sand together at their contact point, Table 25. The UF-nanoSiO_2_ solution provided a compressive strength of more than 2000 psi, but with a reduction in permeability (losses between 4.53% and 11.56%). In addition, they reveal that adding of nanoSiO2 in the urea-formaldehyde causes further growth in the viscosity [67]. Kalgaonkar et al. [121] have studied positively charged modified nanoparticle (5 nm–50 nm) of silica to consolidate a sand formation, Table 25. The sand consolidation contains colloidal silica particles modified, a cationic polymer and a pH modifier. A coated silica particle containing a cationic species non-covalently can bound an outer surface of the particle, to acquire a totally consolidated sand formation. The consolidated mass of sand particles may have strength to hold a pressure load of 700–1000 lbf. Experimental investigation, by check processes, on the modified nanosilica indicated that the consolidated particles are permeable. The check process was conducted as follow, initially, unconsolidated sand was packed in a syringe. After that, the modified nanosilica composition was injected into the sand in the syringe. The treated syringes were put in an oven at 100 °C for 24 h. The treated sand was then removed to the consolidated sand packs. Finally, to check the consolidated packs, two mL of two percent Nacl aqueous solution was injected through the consolidated sand pack. The results revealed that the sand pack has absorbed the NaCl injected solution. The modified nanosilica measurements were conducted at temperature of 24–177 °C [121]. Some advantages and disadvantages of nanoparticles materials methods as reported in literature are summarized in Table 26.

## 15. Conclusions

This article reviews and discusses the application of chemical sand consolidation for controlling sand production from hydrocarbon reservoirs. It recapitulates the characteristics, advantages and disadvantages of different sand consolidation agents, and highlights the improvement trend along their application history. In general, the favorable features of chemical sand consolidation can be summarized as: (i) The applicability for multi-completion wells; (ii) the ability to be carried out in wells having all sizes of perforations, (iii) ability to restrict the migration of fine sand particles completely outside of critical flow region; (iv) applicability for open-hole completions; (v) no mechanical risks associated with screen placement through tubing and in slim hole applications; (vi) does not need any downhole equipment, hence no rig is required and; (vii) more economic than mechanical sand consolidation. The efficiency of a chemical sand consolidation material is typically evaluated, based on its effectiveness in increasing the compressive strength of the targeted rock. However, this is always associated with undesired reduction in rock permeability. Selecting a material that can generate the desired compressive strength, but with the least permeability reduction, is essential. In addition, each of chemical material has its own pros and cons. Thus, the techniques should be selected based on the cost, requirements and reservoir conditions like pressure and temperature.

Many types of polymers have been used to consolidate sand formations such as epoxy, furan, phenol-formaldehyde polymer and polyacrylamide polymer etc. In addition, some types of non-polymeric materials, namely silicate and oxidation, were used occasionally to consolidate loose sand formations at different reservoir conditions. Although most of these sand consolidation agents showed remarkable compressive strength, they suffered from other limitations like reduction in the initial permeability up to 50%, the requirement for high pumping pressure, the requirement for multistage injection, low thermal stability, and multiple phases for consolidating.

In recent years, the technology of chemical sand consolidation application has been improved via enhancing their properties and/or using more innovative materials, such as nanoparticles. Employing nanoparticles in sand control applications has been proven to be promising and an attractive consolidation agent especially at high pressure high temperature conditions. It is concluded that nanoparticles can significantly improve compressive strength with minimal negative impact on the permeability and porosity of sand formation. This review can serve as a screening guide for engineers and scientist to select the chemical method most suitable for sand consolidation of particular unconsolidated formation.

## Figures and Tables

**Figure 1 polymers-12-01069-f001:**
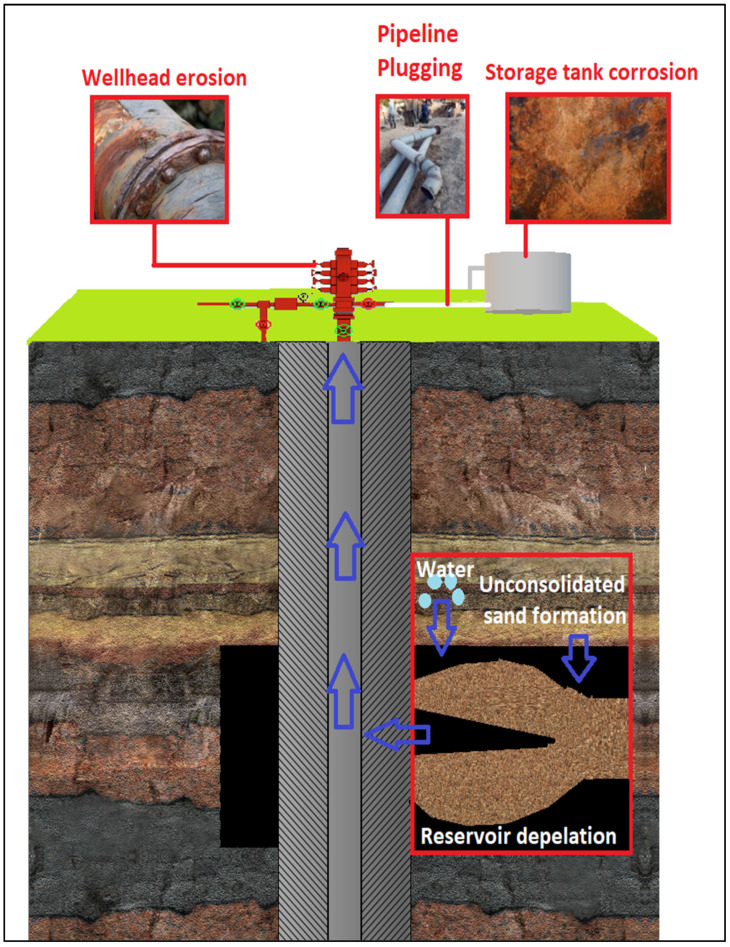
Causes and problems of sand production.

**Figure 2 polymers-12-01069-f002:**
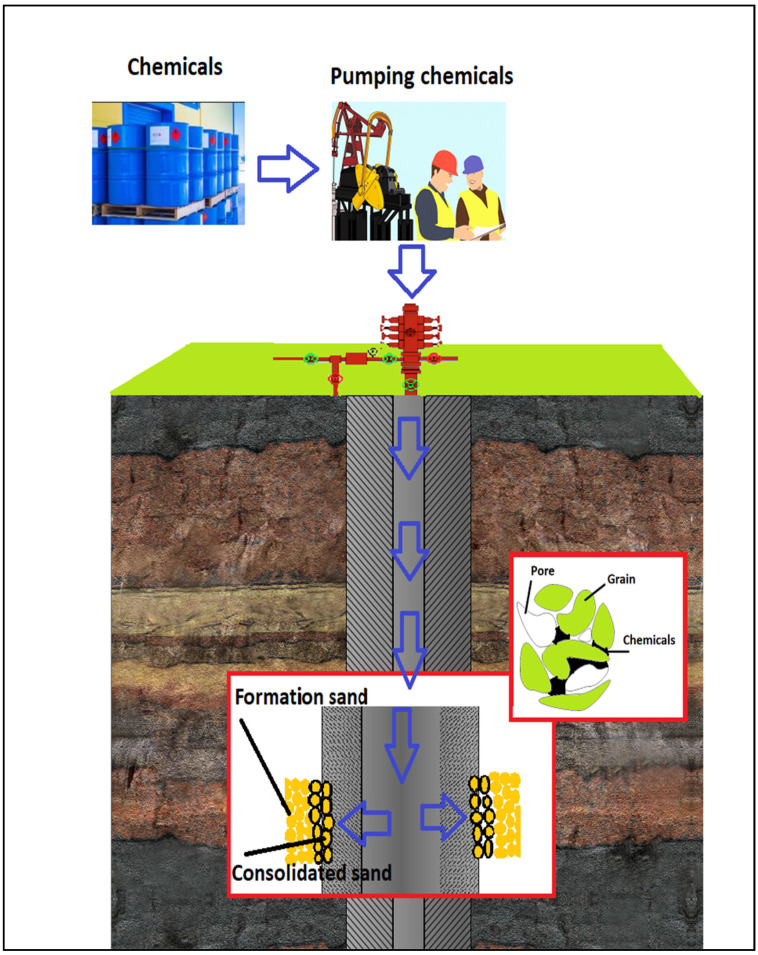
Chemical sand consolidation.

**Figure 3 polymers-12-01069-f003:**
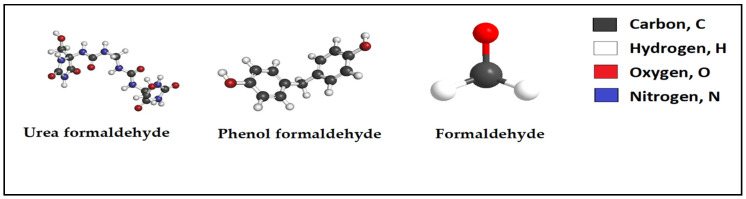
Chemical structure of urea formaldehyde, phenol formaldehyde, and formaldehyde.

**Figure 4 polymers-12-01069-f004:**
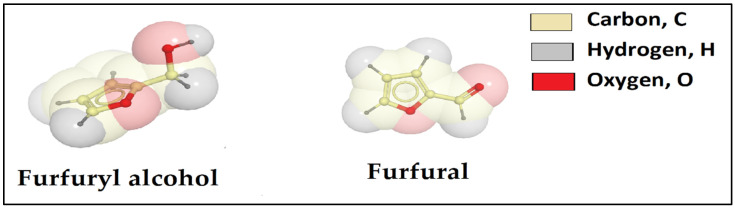
Chemical structure of furfuryl alcohol and furfural.

**Figure 5 polymers-12-01069-f005:**
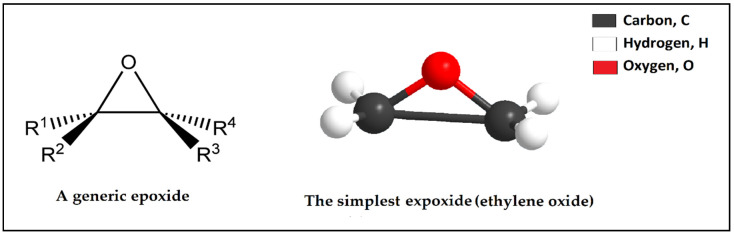
Chemical structure of the epoxide and ethylene oxide.

**Figure 6 polymers-12-01069-f006:**
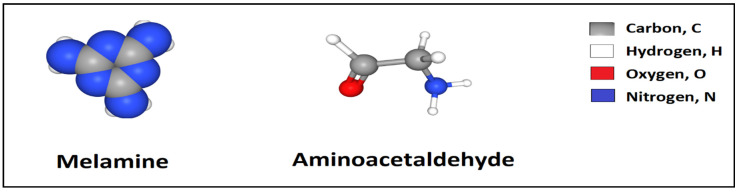
Chemical structure of melamine and aminoacetaldehyde.

**Figure 7 polymers-12-01069-f007:**
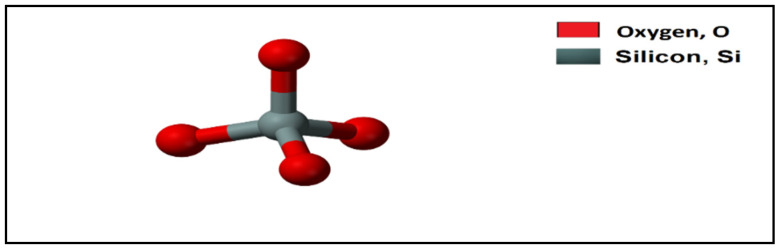
Chemical structure of silicate.

**Figure 8 polymers-12-01069-f008:**
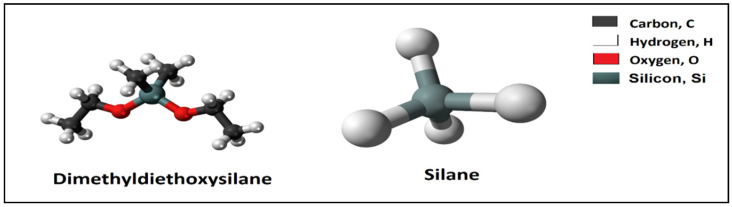
Chemical structure of dimethyldiethoxysilane and silane.

**Figure 9 polymers-12-01069-f009:**
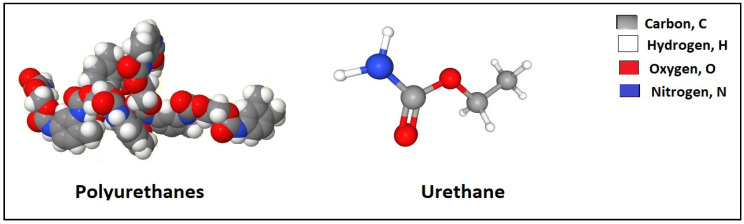
Chemical structure of polyurethanes and urethane.

**Figure 10 polymers-12-01069-f010:**
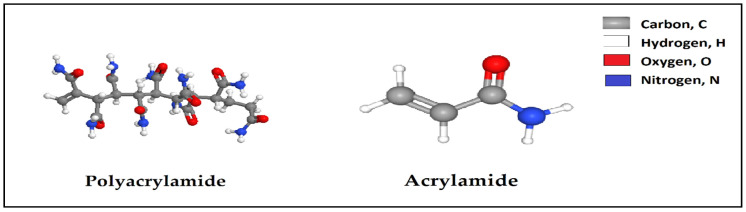
The chemical structure of polyacrylamide and acrylamide.

**Table 1 polymers-12-01069-t001:** Phenol-formaldehyde resin and urea-formaldehyde resin methods to consolidate sand formation.

Reference	Materials	Compressive Strength(psi)	Permeability	Temperature Range (°C)
Spain [30]	plastic (phenol-formaldehyde) resin	1000	original 50%	38–95.5
Talaghat et al. [32]	a modified phenol-formaldehyde resin	3000	1500 mD and 3500 mD	29.5 and 94
Mishra and Ojha [33]	Urea-formaldehyde, potassium silicate and ammonium chloride.	1300	64.65% permeability retention	up to 100

**Table 2 polymers-12-01069-t002:** Advantages and disadvantages of phenol-formaldehyde resin and urea-formaldehyde resin.

Advantages	Disadvantages
High tensile strength.	Decreased the permeability up to 50%.
Flexural modulus.	The issues of placement and reliability.
A high heat distortion temperature.	Short intervals injection.
Low water absorption.	Temperature sensitivity up to 100 °C.
Mold shrinkage.	Difficult to apply, [34].
High surface hardness.	Multiple phases for consolidating.
Elongation at break.	
Volume resistance.	
A refractive index [35].	
Applied in wellbore open.	
No need for screens and liners.	
Increased the compressive strength up to 3000 psi	
More economic than mechanical methods.	
Applied for multi-completion wells.	
Used in wells having all sizes of perforations.	
Control fine sand particles.	
No mechanical risks.	
No need any downhole equipment, so no rig is used, [36].	

**Table 3 polymers-12-01069-t003:** Furan resin methods to control sand formation.

Reference	Materials	Compressive Strength (psi)	Permeability	Temperature Range (°C)
Young [39]	a furan resin (furfuryl alcohol resin)	3000	-	15.5 to 149
Young [40]	UF-85 concentrate, furfuryl alcohol, urea	1072	-	27, 60, 65.5, 177
Young [41]	terahydrofurfuryl methacrylate and a furan resin	1170	-	72
Friedman et al. [21]	polymerizable monomer or oligomer	-	-	30, 60
Fader et al. [42]	a modified furan resin	550	permeability reduction 10%	15.5–160
Parlar et al. [43]	furan resin	2000	average permeability (80–85)%	38, 66, 93, 121
Todd et al. [44]	furan resin, an aminosilane resin-to-particulate	3000	-	93
Appah [45]	furan-phenolic resin	3000	-	-
Nguyen [46]	a furfuryl alcohol resin, furfuryl alcohol	80–1642	-	27 and less than 149

**Table 4 polymers-12-01069-t004:** Advantages and disadvantages of furan resin.

Advantages	Disadvantages
Good corrosion resistance to the inorganic acid.	The placement of the chemical through all perforations is critical to success, [43].
Increased the compressive strength up to 3000 psi.	The low flash point of the resin
Good heat resistance.	High required pumps [39].
Furan resins belong to thermosetting resin and could cross-link with each other and cure when heated without adding the curing agent.	Poor oxidation resistance of furan resin [47].
Furan resins are good miscible with a myriad of thermosetting resins, thus a lot of products with diverse performance can be formed [47].	The multistage process injection.
The solid coupling agent can be used to break separating films of viscous carrier fluid between resin coated particulate solids and the surface active agent which can be used to cause the resin composition to flow to the contact points [44].	The weaknesses of this method are that the unfriendly material like sulfuric acid are used in the consolidation solution [21].

**Table 5 polymers-12-01069-t005:** Epoxy resin methods to consolidate sand formation.

Reference	Materials	Compressive Strength (psi)	Permeability	Temperature Range (°C)
Penberthy et al. [52]	an epoxy resin, a brine preflush and solvent preflush	2000	-	-
Dees et al. [53]	an epoxy resin, an explosive, a gas generator and a particulate diverting agent.	5000	a permeability to original 67%	38 to 93
Dees [54]	an epoxy resin and a gas generator	7000	a permeability to original 50%	10 to 121
Dewprashad et al. [55]	a new HT epoxy resin	1340	-	up to 204
Chaloupka et al. [56]	a catalyzed epoxy	-	63% regained permeability	93–177
Marfo et al. [13]	an epoxy resin	600–700	a permeability to original (60–90)%	38–107
Riyanto et al. [57]	epoxy resin, coupling agent, accelerator and hardener	-	-	63
Hadi et al. [58]	an epoxy resin	614–1816	(100–1068) md	38.9–79.4
Palencia et al. [14]	an epoxy resin	217.21–694	8.15 × 10^−7^–1095.1 mD	62
Alanqari et al. [59]	1- Bisphenol A/Epichorohydrin and Butyl glycidyl ether and cyclohxanedimethanol resins cured with Polyoxypropylene Diamine. 2- Diglycidylether of bisphenol-A epoxy resin.	-	-	1- 120.5 2- 146
Al-Mulhem [60]	an epoxy resin and curing agent (Expedite A& B by Halliburton Energy Services)	-	a permeability to original 70%	90
Eluru and Salla. [61]	a sugar based hardenable resin (glucofuranoside based trifunctional epoxy resin and glucofuranoside based trifunctional epoxy resin) and a hardening agent	-	-	160

**Table 6 polymers-12-01069-t006:** Advantages and disadvantages of epoxy resin.

Advantages	Disadvantages
This method does not effect on fluid pH and has less impact on gel breakers [55].	High required pumps [39].
The resin treatment takes adequate time to become hardened because of a lower cure rate at high temperature, so the consolidation operation can become easier to palace the resin at the target sand zone [55].	Reduction of the original permeability up to 50%.
Substantial mechanical strength [62].	The multistage process injection.
High resistance to heat.	Irritant to the eyes and skin [50]
Outstanding resistance to chemical and substantial adhesive strength [48]	
Capable with higher temperatures up to (215 °C) [51].	
Increased the compressive strength up to 7000 psi.	
Low cost.	
Effective electrical insulation.	
Good adhesion.	
High performance composites.	
Electronic packaging materials.	
Versatile processability and adhesive properties [63].	
Protective coatings [62].	

**Table 7 polymers-12-01069-t007:** Amino-aldehyde polymer methods to consolidate sand formation.

Reference	Materials	Compressive Strength (psi)	Permeability	Temperature Range (°C)
Larsen et al. [65]	Quasi Natural Consolidation (QNC) Ca2+, urease and urea	1531	permeability decreased about 25% from the original	25–65
Lahalih and Ghloum [66]	amino-aldehyde resins	497–1422	-	(60, 80, 110)

**Table 8 polymers-12-01069-t008:** Advantages and disadvantages of amino-aldehyde polymer.

Advantages	Disadvantages
Outstanding thermodynamic and mechanical properties.	Expensive [68].
Not toxic.	Decreased the permeability up to 25%.
More compatible with the fluid in the formation [69].	The issues of placement and reliability.
Melamine formaldehyde is stain-resistant and resistant to strong solvents and water.	Short intervals injection.
Applied in wellbore open.	Decreased the permeability up to 25%.
No need for screens and liners.	Temperature sensitivity up to 110 °C.
More economic than mechanical methods.	Difficult to apply, [34].
Applied for multi-completion wells.	Multiple phases for consolidating.
Used in wells having all sizes of perforations.	Increased the compressive strength only up to 1531 psi.
Control fine sand particles.	Ingestion leads to kidney failure [70].
No mechanical risks.	
No need any downhole equipment, so no rig is used, [36].	

**Table 9 polymers-12-01069-t009:** Silicate polymer methods to consolidate sand formation.

Reference	Materials	Compressive Strength (psi)	Permeability	Temperature Range (°C)
Anthony [74]	organic silicate	-	-	-
Shu [75]	an aqueous silicate solution	-	-	79.5
Dwivedi and Singh [76]	sodium silicate solution	0–398	-	20, 40, 60
Cobianco et al. [77]	sodium silicate solution	284–427	a permeability retention was (70–80)%	60

**Table 10 polymers-12-01069-t010:** Advantages and disadvantages of silicate polymer.

Advantages	Disadvantages
Safer and less sensitive to formation and sand fluid, to avoid the problems of safety issues and the negative effect on the formation [74].	Increased the compressive strength only up to 427 psi.
The retention permeability is high without given a specific value of permeability retention [75].	The issues of placement and reliability.
Adding silicates decrease the viscosity of slurries.	Short intervals injection.
Adding silicates decrease firing time, sinter shrinkage.	Temperature sensitivity up to 79.5 °C.
Increases the strength of the materials.	Multiple phases for consolidating.
Silicates rise temperature and acid resistance [78].	

**Table 11 polymers-12-01069-t011:** Oxidation and hydrocarbon methods to consolidate sand formation.

Reference	Materials	Compressive Strength (psi)	Permeability	Temperature Range (°C)
Jennings et al. [80]	oxygenated foam	3000	a permeability to the original 70%	138
Aggour et al. [83]	low-temperature oxidation (LTO) of crude oil	375–1264	a permeability loss was 22%	100–150
Aggour et al. [85]	the developed LTO	1800–2300	retained permeability to the original (86.4–95.5%)	100–150
Khamatnurova [86]	a long chain hydrocarbon viscosifier, a curing agent and a thiol crosslinking agent	>300	-	60

**Table 12 polymers-12-01069-t012:** Advantages and disadvantages of oxidation and hydrocarbon methods.

Advantages	Disadvantages
Increased the compressive strength up to 3000 psi.	The issues of placement and reliability.
Applied in opened wellbore.	Short intervals injection.
Without screens and liners.	Temperature sensitivity up to 150 °C.
More economic than mechanical methods.	
Applied for multi-completion wells.	
Used in wells having all sizes of perforations.	
Control fine sand particles.	
No mechanical risks.	
No need any downhole equipment, so no rig is used, [36].	

**Table 13 polymers-12-01069-t013:** Shape memory polymers methods to consolidate sand formation.

Reference	Materials	Compressive Strength (psi)	Permeability (D)	Temperature Range (°C)
Carrejo et al. [25]	Shape Memory Polymer A pore throat (60–160) μm. Less than 44 μm to pass through.	-	80	60
Wang and Osunjaye [90]	Shape Memory Polymer	-	Over 30	4.4, (37.8–93.3)
Leung et al. [91], Fuxa et al. [92]	Shape Memory Polymer		0.005–0.6	60

**Table 14 polymers-12-01069-t014:** Advantages and disadvantages of shape memory polymers methods.

Advantages	Disadvantages
High elastic deformation.	Stop the sand particles greater than 43 microns to control sand production.
Low cost.	Temperature sensitivity up to 93 °C.
Low density.	
Low biocompatibility.	
Low biodegradability.	
Capable of recovering under low deformation level [93].	

**Table 15 polymers-12-01069-t015:** Hydrolysate or precondensate consolidation agent method to consolidate sand formation.

Reference	Materials	Compressive Strength (psi)	Permeability	Temperature (°C)
Endres et al. [95]	a hydrolysate	335	-	150

**Table 16 polymers-12-01069-t016:** Advantages and disadvantages of hydrolysate or precondensate consolidation agent method.

Advantages	Disadvantages
Silane coupling agents are compounds whose molecules comprise function groups that bond with organic and inorganic materials.	Increased the compressive strength only up to 335 psi.
Silane coupling agents are useful to improve the mechanical strength of composite materials.	The issues of placement and reliability.
Silane coupling agents improve adhesion.	Short intervals injection.
Heat resistance.	Temperature sensitivity up to 150 °C.
High cross-linking [96].	

**Table 17 polymers-12-01069-t017:** Permeability enhancing additive (PEA) method to consolidate sand formation.

Reference	Materials	Compressive Strength (psi)	Permeability	Temperature (°C)
Kalgaonkar et al. [97]	a permeability enhancing additive (PEA)	1000	70% regain permeability	90

**Table 18 polymers-12-01069-t018:** Advantages and disadvantages of permeability enhancing additive (PEA) method.

Advantages	Disadvantages
Decrease the viscosity of the resin to be easily pumping into the wellbore.	Increased the compressive strength only up to 1000 psi.
Can be used as the liquid phase to fill in the pore spaces of the consolidation sand formation to prevent the permeability of the formation.	Short intervals injection.
Easy in term of mixing procedure.	Temperature sensitivity up to 90 °C
Maintains the permeability.	
Can give a controllable curing time up to some days, so avoiding any premature setting of the resin in the wellbore [97].	

**Table 19 polymers-12-01069-t019:** Polyurethane resins methods to consolidate sand formation.

Reference	Materials	Compressive Strength	Permeability	Temperature (°C)
Spurlock et al. [100]	kerosene or diesel oil, polyurethane solution.	161–600 psi	a permeability to original 80%	60
Liu et al. [24,101]	Poly-oxypropylene diol, poly-oxyethylene glycol, poly-caprolaclone glycol, and toluene.	Shear strength (cohesion values) (120.19, 140.28, 204.22) kPa	-	20

**Table 20 polymers-12-01069-t020:** Advantages and disadvantages of polyurethane resins methods.

Advantages	Disadvantages
Important adhesion to base materials, [98,99].	Increased the compressive strength only up to 600 psi.
High resistance to weathering.	The issues of placement and reliability.
High resistance to solvents.	Short intervals injection.
High resistance to mechanical damage [102].	Temperature sensitivity up to 60 °C.
Elastomers, sealants and elastoplastics [103].	Multiple phases for consolidating.
High performance adhesives.	
High amounts of crosslinking provide rigid polymers	
Thermosetting polymer [104].	

**Table 21 polymers-12-01069-t021:** Polyacrylamide polymer methods to consolidate sand formation.

Reference	Materials	Compressive Strength	Permeability	Temperature Range (°C)
Falk [106]	aqueous solution comprises an acrylamide polymer	-	-	(50–100)
Sydansk [107]	an acrylamide-polymer	-	-	up to 127
Salehi et al. [108]	polymer hydrolyzed acrylamido propyl sulfonated acid and a cross-linker chromium triacetate	30 times the original one	-	90

**Table 22 polymers-12-01069-t022:** Advantages and disadvantages of polyacrylamide polymer method.

Advantages	Disadvantages
Improves a sand consolidation with high compressive strength 30 times the original one.	The issues of placement and reliability.
Has adjustable viscosity [108].	Short intervals injection.
Flocculate solids in a liquid [109].	
Has thermal stability.	Multiple phases for consolidating.
Has good mechanical strength.	
Water-soluble polymer.	
Has high ionic conductivity [110].	
Polyacrylamide polymers have effects of filming and preventing scale [111].	
Polyacrylamide polymers can be cross-linked.	
Polyacrylamide polymers can be drag-reducing agent [112].	
Polyacrylamide polymers can be water treatment agent [113].	

**Table 23 polymers-12-01069-t023:** Water-based and saline-based methods to consolidate sand formation.

Reference	Materials	Compressive Strength (psi)	Permeability	Temperature (°C)
Bhasker et al. [114]	an aqueous based epoxy resin	-	regained permeability 80%	27–113
Songire et al. [115]	an aqueous based epoxy resin	-	-	-
Othman et al. [116]	a solvent-based epoxy resin	1310	regained permeability 87%	85
Shang et al. [69]	a melamine formaldehyde resin water-based	742–911	932 mD to 2736 mD	60
Reddy et al. [117]	a polyvalent metal salt of a carboxylic water-based	647	-	100
George et al. [118]	saline-based	200	regained permeability 80%	-

**Table 24 polymers-12-01069-t024:** Advantages and disadvantages of water-based and saline-based methods.

Advantages	Disadvantages
Safer.	Increased the compressive strength only up to 200 psi.
Environmentally friendly.	Temperature sensitivity up to 113 °C.
Low viscos (1 cP) (i.e., requiring less pumping pressure).	
A value of regained permeability is (more than 80%)	
Applied in wellbore open.	
No need for screens and liners.	
More economic than mechanical methods [36]	

**Table 25 polymers-12-01069-t025:** Nanoparticles materials methods to consolidate sand formation.

Reference	Materials	Compressive Strength	Permeability	Temperature Range (°C)
Espin et al. [120]	hydroxyls and inorganic like SiO2 (1–200 nm)	Young’s Modulus 1 × 10^6^ psi	-	-
Mishra and Ojha [67]	silicon dioxide nanoparticles with urea formaldehyde resin [UF]	2000 psi	permeability losses between 4.53% and 11.56%	60–160
Kalgaonkar and Fakuen [121]	positively charged modified particles of nanosilica	hold a pressure load (700–1000) lbf	-	(24–177)

**Table 26 polymers-12-01069-t026:** Advantages and disadvantages of nanoparticles materials methods.

Advantages	Disadvantages
Nanoparticles used as a displacing fluid to dislodge formation fluids into unconsolidated formation and away from the well [120].	Temperature sensitivity up to 177 °C.
Nanoparticles can form a monolayer of consolidating material across the loose sand to control the sand formation with desired permeability characteristics. The monolayer can cement the sand grains together and ensures a retained permeability through the treatment material to facilitate production of hydrocarbons.	
The nanoparticles materials can sigificately improve formation strength without negatively impacting the permeability and porosity [121].

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
