# Peer review of "Chemical Sand Consolidation: From Polymers to Nanoparticles"

_polymers, 2020, doi:10.3390/polym12051069_

Round 1

Reviewer 1 Report

The authors have addressed all the problems of their previous submission in this revised version. I recommend the publication of the manuscript.

Reviewer 2 Report

This manuscript has been improved by the authors. It can be accepted for publication.

This manuscript is a resubmission of an earlier submission. The following is a list of the peer review reports and author responses from that submission.

Round 1

Reviewer 1 Report

The work by Alakbari et al. presents a detailed study about different processes for sand consolidation based in the use of polymer derivates and other derivates. Even though the work is detailed and analyzes carefully, even though the style is a little difficult to follow, the different aspects involved in the technological impact of the process, the polymers are not the center of interest of the work, with the engineering aspects focusing the study. Thus, my recommendation is to submit the work to a journal more focused in engineering processes, maybe Processes. Just in case that authors decide to resubmit the manuscript to Polymers some comments:

-Abstract provide a confuse description which does not make it clear the interest of the current manuscript.

-The introduction does not explain clearly the importance of this work from a perspective of the polymer use.

-The importance of the polymers used should be stated within the manuscript a no only the industrial or technological interest of the process should be described.

Reviewer 2 Report

The paper deals with ‘“Chemical Sand Consolidation from Polymers to Nanoparticles”. This review article can be an interesting and scientifically relevant to publish in Polymers. However, few alterations could be made.

  1. Please check the spelling. The format, front should be kept constant through the manuscript.
  2. It is difficult to see what the difference is between this review and the literature. Therefore, the authors could provide more discussion to get insight on the originality of this review paper.

We recommend reading the following paper:

Saurabh Mishra, Keka Ojha. Chemical Sand Consolidation: An Overview. Journal of Petroleum Engineering & Technology, ISSN: 2231-1785(online), ISSN: 2321-5178(print), Volume 5, Issue 2.

  1. Can author correct the errors for the references, for instance: line 160, line 372, line 441, line 502.
  2. Authors discussed urea formaldehyde in part 2, and part 5. Is it better to move them together?
  3. If possible, authors can summary the advantages and disadvantages of using the techniques (polymer, PEA, Oxidation, hydrolysate or precondensate consolidation agent etc.) in a Table. This should include the characteristics of different sand consolidation agents and highlights the improvement trend along the history of cholidation.
  4. What is the significance and utility of this work that should be addressed concisely in the conclusion?
  5. Finally, please do a thorough editing effort in the English language and make sure that spaces and proper grammar rules are followed throughout the manuscript.

Reviewer 3 Report

The review article summarized and discussed the application of chemical methods, mainly polymers, to control sand production in hydrocarbon reservoirs, an interesting application of polymers. I recommend the publication in this journal after the authors addressed the following comments.

  1. It would be nice to add a scheme or picture to describe a general process using polymer to consolidate the sand in the reservoir. It would be easier for readers from other fields to understand such process.
  2. Similarly, please add some schemes or pictures when you discuss the cited examples.
  3. There are some typos or formatting errors.

Reviewer 4 Report

This manuscript presents an up-to-date overview of the chemical sand consolidation techniques by highlighting and deliberating the most popular chemical methods and the latest advanced approaches. Authors illustrated rich examples and made discussions on them. However, some issues still need to be well addressed.

  1. Can the authors supply a comparison diagram or table to describe the advantages and disadvantages of the chemical sand consolidation methods?
  2. Nearly 60% of references were published earlier than 2010. Can the authors review more new articles in the last decade?
  3. The conclusion is a bit long. Please simplify it.
  4. There are some misused words. Please carefully check the manuscript. For example, in page 4, line 160-161, “Table 1Error! Reference source not found” is not correct.